# ODE$^2$VAE: Deep generative second order ODEs with Bayesian neural networks

**Çağatay Yıldız[1], Markus Heinonen[1,2], Harri Lähdesmäki[1]**
Department of Computer Science
Aalto University, Finland, FI-00076
{cagatay.yildiz, markus.o.heinonen, harri.lahdesmaki}@aalto.fi

## Abstract

We present Ordinary Differential Equation Variational Auto-Encoder (ODE$^2$VAE), a latent second order ODE model for high-dimensional sequential data. Leveraging the advances in deep generative models, ODE$^2$VAE can simultaneously learn the embedding of high dimensional trajectories and infer arbitrarily complex continuous-time latent dynamics. Our model explicitly decomposes the latent space into momentum and position components and solves a second order ODE system, which is in contrast to recurrent neural network (RNN) based time series models and recently proposed black-box ODE techniques. In order to account for uncertainty, we propose probabilistic latent ODE dynamics parameterized by deep Bayesian neural networks. We demonstrate our approach on motion capture, image rotation and bouncing balls datasets. We achieve state-of-the-art performance in long term motion prediction and imputation tasks.

## 1 Introduction

Representation learning has always been one of the most prominent problems in machine learning. Leveraging the advances in deep learning, variational auto-encoders (VAEs) have recently been applied to several challenging datasets to extract meaningful representations. Various extensions to vanilla VAE have achieved state-of-the-art performance in hierarchical organization of latent spaces, disentanglement and semi-supervised learning (Tschannen et al., 2018).

VAE based techniques usually assume a static data, in which each data item is associated with a single latent code. Hence, auto-encoder models for sequential data have been overlooked. More recently, there have been attempts to use recurrent neural network (RNN) encoders and decoders for tasks such as representation learning, classification and forecasting (Srivastava et al., 2015; Lotter et al., 2016; Hsu et al., 2017; Li and Mandt, 2018). Other than neural ordinary differential equations (ODEs) (Chen et al., 2018b) and Gaussian process prior VAEs (GPPVAE) (Casale et al., 2018), aforementioned methods operate in discrete-time, which is in contrast to most of the real-world datasets, and fail to produce plausible long-term forecasts (Karl et al., 2016).

In this paper, we propose ODE$^2$VAEs that extend VAEs for sequential data with a latent space governed by a continuous-time probabilistic ODE. We propose a powerful second order ODE that allows modelling the latent dynamic ODE state decomposed as position and momentum. To handle uncertainty in dynamics and avoid overfitting, we parameterise our latent continuous-time dynamics with deep Bayesian neural networks and optimize the model using variational inference. We show state-of-the-art performance in learning, reproducing and forecasting high-dimensional sequential systems, such as image sequences. An implementation of our experiments and generated video sequences are provided at https://github.com/cagatayyildiz/ODE2VAE.

## 2 Probabilistic second-order ODEs

We tackle the problem of learning low-rank latent representations of possibly high-dimensional sequential data trajectories. We assume data sequences $\mathbf{x}_{0:N} := (\mathbf{x}_0, \mathbf{x}_1, \ldots, \mathbf{x}_N)$ with individual *frames* $\mathbf{x}_k \in \mathbb{R}^D$ observed at time points $t_0, \ldots, t_N$. We will present the methodology for a single data sequence $\mathbf{x}_{0:N}$ for notational simplicity, but it is straighforward to extend our method to multiple sequences. The observations are often at discrete spacings, such as individual images in a video sequence, but our model also generalizes to irregular sampling.

We assume that there exists an underlying generative low-dimensional continuous-time dynamical system, which we aim to uncover. Our goal is to learn latent representations $\mathbf{z}_t \in \mathbb{R}^d$ of the sequence dynamics with $d \ll D$, and reconstruct observations $\mathbf{x}_t \in \mathbb{R}^D$ for missing frame imputation and forecasting the system past observed time $t_N$.

### 2.1 Ordinary differential equations

In discrete-time sequential systems the state *sequence* $\mathbf{z}_0, \mathbf{z}_1, \ldots$ is indexed by a discrete variable $k \in \mathbb{Z}$, and the state progression is governed by a transition function on the change $\Delta \mathbf{z}_k = \mathbf{z}_k - \mathbf{z}_{k-1}$. Examples of such models are auto-regressive models, Markov chains, recurrent models and neural network layers.

In contrast, continuous-time sequential systems model the state *function* $\mathbf{z}_t : \mathcal{T} \to \mathbb{R}^d$ of a continuous, real-valued time variable $t \in \mathcal{T} = \mathbb{R}$. The state evolution is governed by a first-order time derivative

$$\dot{\mathbf{z}}_t := \frac{d\mathbf{z}_t}{dt} = \mathbf{h}(\mathbf{z}_t), \tag{1}$$

that drives the system state forward in infinitesimal steps over time. The differential $\mathbf{h} : \mathbb{R}^d \to \mathbb{R}^d$ induces a *differential field* that covers the input space. Given an initial location vector $\mathbf{z}_0 \in \mathbb{R}^d$, the system then follows an *ordinary differential equation* (ODE) model with state solutions

$$\mathbf{z}_T = \mathbf{z}_0 + \int_0^T \mathbf{h}(\mathbf{z}_t)dt. \tag{2}$$

The state solutions are in practise computed by solving this initial value problem with efficient numericals solvers, such as Runge-Kutta (Schober et al., 2019). Recently several works have proposed learning ODE systems $\mathbf{h}$ parametrised as neural networks (Chen et al., 2018b) or as Gaussian processes (Heinonen et al., 2018).

### 2.2 Bayesian second-order ODEs

First-order ODEs are incapable of modelling high-order dynamics[1], such as acceleration or the motion of a pendulum. Furthermore, ODEs are deterministic systems unable to account for uncertainties in the dynamics. We tackle both issues by introducing Bayesian neural second-order ODEs

$$\ddot{\mathbf{z}}_t := \frac{d^2\mathbf{z}_t}{d^2t} = \mathbf{f}_{\mathcal{W}}(\mathbf{z}_t, \dot{\mathbf{z}}_t), \tag{3}$$

which can be reduced to an equivalent system of two coupled first-order ODEs

$$\begin{cases} \dot{\mathbf{s}}_t &= \mathbf{v}_t \\ \dot{\mathbf{v}}_t &= \mathbf{f}_{\mathcal{W}}(\mathbf{s}_t, \mathbf{v}_t) \end{cases}, \qquad \begin{bmatrix} \mathbf{s}_T \\ \mathbf{v}_T \end{bmatrix} = \begin{bmatrix} \mathbf{s}_0 \\ \mathbf{v}_0 \end{bmatrix} + \int_0^T \underbrace{\begin{bmatrix} \mathbf{v}_t \\ \mathbf{f}_{\mathcal{W}}(\mathbf{s}_t, \mathbf{v}_t) \end{bmatrix}}_{\tilde{\mathbf{f}}_{\mathcal{W}}(\mathbf{z}_t)} dt, \tag{4}$$

where (with a slight abuse of notation) the state tuple $\mathbf{z}_t = (\mathbf{s}_t, \mathbf{v}_t)$ decomposes into the state *position* $\mathbf{s}_t$, which follows the state *velocity* (momentum) $\mathbf{v}_t$. The velocity or evolution of change is governed by a neural network $\mathbf{f}_{\mathcal{W}}(\mathbf{s}_t, \mathbf{v}_t)$ with a collection of weight parameters $\mathcal{W} = \{\mathbf{W}_\ell\}_{\ell=1}^L$ over its $L$ layers and the bias terms. We assume a prior $p(\mathcal{W})$ on the weights resulting in a Bayesian neural network (BNN). Each weight sample, in turn, results in a deterministic ODE trajectory (see Fig. 1).

The BNN *acceleration field* $\mathbf{f}_{\mathcal{W}} : \mathbb{R}^d \times \mathbb{R}^d \to \mathbb{R}^d$ depends on both state and velocity. For instance, in a pendulum system the acceleration $\ddot{\mathbf{z}}$ depends on both its current location and velocity. The system is now driven forward from starting position $\mathbf{s}_0$ and velocity $\mathbf{v}_0$, with the BNN determining only how the velocity $\mathbf{v}_t$ evolves.

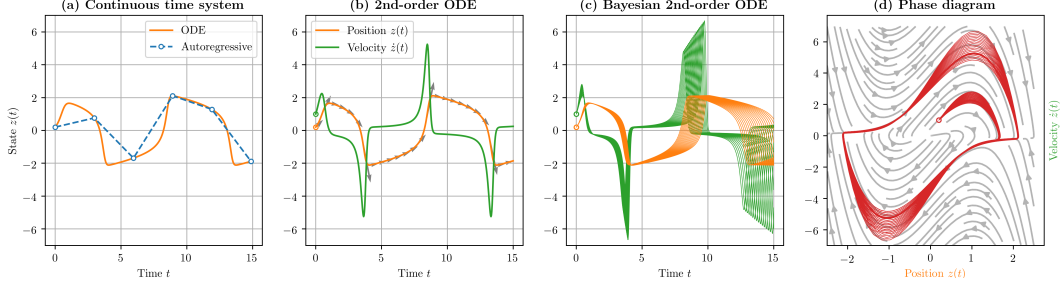

Figure 1: Illustration of dynamical systems. A continuous-time system underlying a discrete-time model **(a)** can be extended to a 2nd-order ODE with velocity component **(b)**. A Bayesian ODE characterises uncertain differential dynamics **(c)**, with the corresponding position-velocity phase diagram **(d)**. The gray arrows in **(d)** indicate the BNN $\mathbf{f}_{\mathcal{W}}(\mathbf{s}_t, \mathbf{v}_t)$ mean field wrt $p(\mathcal{W})$.

## 2.3 Second order ODE flow

The ODE systems are denoted as continuous normalizing flows when they are applied on random variables $\mathbf{z}_t$ (Rezende et al., 2014; Chen et al., 2018a; Grathwohl et al., 2018). This allows following the progression of its density through the ODE. Using the instantaneous change of variable theorem (Chen et al., 2018a), we obtain the instantaneous change of variable for our second order ODEs as

$$\frac{\partial \log q(\mathbf{z}_t | \mathcal{W})}{\partial t} = -\mathrm{Tr}\left( \frac{d\tilde{\mathbf{f}}_{\mathcal{W}}(\mathbf{z}_t)}{d\mathbf{z}_t} \right) dt = -\mathrm{Tr}\left( \begin{matrix} \frac{\partial \mathbf{v}_t}{\partial \mathbf{s}_t} & \frac{\partial \mathbf{v}_t}{\partial \mathbf{v}_t} \\ \frac{\partial \mathbf{f}_{\mathcal{W}}(\mathbf{s}_t, \mathbf{v}_t)}{\partial \mathbf{s}_t} & \frac{\partial \mathbf{f}_{\mathcal{W}}(\mathbf{s}_t, \mathbf{v}_t)}{\partial \mathbf{v}_t} \end{matrix} \right) = -\mathrm{Tr}\left( \frac{\partial \mathbf{f}_{\mathcal{W}}(\mathbf{s}_t, \mathbf{v}_t)}{\partial \mathbf{v}_t} \right), \tag{5}$$

which results in the log densities over time,

$$\log q(\mathbf{z}_T | \mathcal{W}) = \log q(\mathbf{z}_0 | \mathcal{W}) - \int_0^T \mathrm{Tr}\left( \frac{\partial \mathbf{f}_{\mathcal{W}}(\mathbf{s}_t, \mathbf{v}_t)}{\partial \mathbf{v}_t} \right) dt. \tag{6}$$

# 3 ODE$^2$VAE model

In this section we propose a novel dynamic VAE formalism for sequential data by introducing a second order Bayesian neural ODE model in the latent space to model the data dynamics. We start by reviewing the standard VAE models and then extend it to our ODE$^2$VAE model.

With auto-encoders, we aim to learn latent representations $\mathbf{z} \in \mathbb{R}^d$ for complex observations $\mathbf{x} \in \mathbb{R}^D$ parameterised by $\theta$, where often $d \ll D$. The posterior $p_\theta(\mathbf{z}|\mathbf{x}) \propto p_\theta(\mathbf{x}|\mathbf{z})p(\mathbf{z})$ is proportional to the prior $p(\mathbf{z})$ of the latent variable and the *decoding* likelihood $p_\theta(\mathbf{x}|\mathbf{z})$. Parameters $\theta$ could be optimized by maximizing the marginal log likelihood but that generally involves intractable integrals. In variational auto-encoders (VAE) an amortized variational approximation $q_\phi(\mathbf{z}|\mathbf{x}) \approx p_\theta(\mathbf{z}|\mathbf{x})$ with parameters $\phi$ is used instead (Jordan et al., 1999; Kingma and Welling, 2013; Rezende et al., 2014). Variational inference that minimizes the Kullback-Leibler divergence, or equivalently maximizes the evidence lower bound (ELBO), results in efficient inference.

## 3.1 Dynamic model

Building upon the ideas from black-box ODEs and variational auto-encoders, we propose to infer continuous-time latent position and velocity trajectories that live in a much lower dimensional space but still match the data well (see Fig. 2 for illustration). For this, consider a generative model that consists of three components: *(i)* a distribution for the initial position $p(\mathbf{s}_0)$ and velocity $p(\mathbf{v}_0)$ in the latent space , *(ii)* true (unknown) dynamics defined by an acceleration field, and *(iii)* a *decoding* likelihood $p(\mathbf{x}_i|\mathbf{s}_i)$.

$$\mathbf{s}_0 \sim p(\mathbf{s}_0) \tag{7}$$

$$\mathbf{v}_0 \sim p(\mathbf{v}_0) \tag{8}$$

$$\mathbf{s}_t = \mathbf{s}_0 + \int_0^t \mathbf{v}_\tau d\tau \tag{9}$$

$$\mathbf{v}_t = \mathbf{v}_0 + \int_0^t \mathbf{f}_{\text{true}}(\mathbf{s}_\tau, \mathbf{v}_\tau) d\tau \tag{10}$$

$$\mathbf{x}_i \sim p(\mathbf{x}_i|\mathbf{s}_i) \quad i \in [0, N] \tag{11}$$

The generative model is given in Eqs. 7-11. Note that the decoding likelihood is defined only from the position variable. Velocity thus serves as an auxiliary variable, driving the position forward.

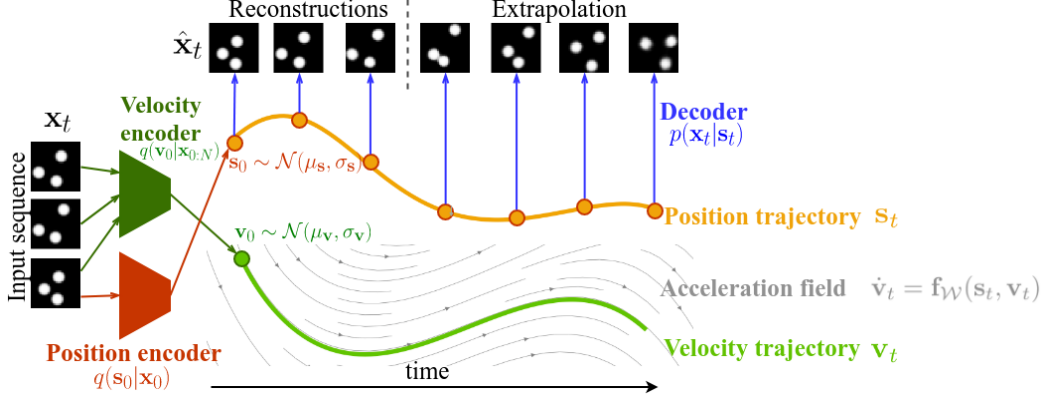

Figure 2: A schematic illustration of ODE$^2$VAE model. Position encoder ($\boldsymbol{\mu}_\mathbf{s}, \boldsymbol{\sigma}_\mathbf{s}$) maps the first item $\mathbf{x}_0$ of a high-dimensional data sequence into a distribution of the initial position $\mathbf{s}_0$ in a latent space. Velocity encoder ($\boldsymbol{\mu}_\mathbf{v}, \boldsymbol{\sigma}_\mathbf{v}$) maps the first $m$ high-dimensional data items $\mathbf{x}_{0:m}$ into a distribution of the initial velocity $\mathbf{v}_0$ in a latent space. Probabilistic latent dynamics are implemented by a second order ODE model $\tilde{\mathbf{f}}_\mathcal{W}$ parameterised by a Bayesian deep neural network ($\mathcal{W}$). Data points in the original data domain are reconstructed by a decoder.

## 3.2 Variational inference

As with standard auto-encoders, optimization of ODE$^2$VAE model parameters with respect to marginal likelihood would result in intractability and thus we resort to variational inference (see Fig. 2). We first combine the latent position and velocity components into a single vector $\mathbf{z}_t := (\mathbf{s}_t, \mathbf{v}_t)$ for notational clarity, and assume the following factorized variational approximation for the unobserved quantities $q(\mathcal{W}, \mathbf{z}_{0:N}|\mathbf{x}_{0:N}) = q(\mathcal{W})q_{\text{enc}}(\mathbf{z}_0|\mathbf{x}_{0:N})q_{\text{ode}}(\mathbf{z}_{1:N}|\mathbf{x}_{0:N}, \mathbf{z}_0, \mathcal{W})$. As decribed in subsection 2.2, true dynamics are approximated by a BNN parameterized by $\mathcal{W}$ with the following variational approximation: $q(\mathcal{W}) = \mathcal{N}(\mathcal{W}|\mathbf{m}, s\mathbf{I})$. We use an amortized variational approximation for the latent initial position and velocity

$$q_{\text{enc}}(\mathbf{z}_0|\mathbf{x}_{0:N}) = q_{\text{enc}}\left(\begin{pmatrix}\mathbf{s}_0\\\mathbf{v}_0\end{pmatrix}\Bigg|\,\mathbf{x}_{0:N}\right) = \mathcal{N}\left(\begin{pmatrix}\boldsymbol{\mu}_\mathbf{s}(\mathbf{x}_0)\\\boldsymbol{\mu}_\mathbf{v}(\mathbf{x}_{0:m})\end{pmatrix}, \begin{pmatrix}\text{diag}(\boldsymbol{\sigma}_\mathbf{s}(\mathbf{x}_0)) & \mathbf{0}\\\mathbf{0} & \text{diag}(\boldsymbol{\sigma}_\mathbf{v}(\mathbf{x}_{0:m}))\end{pmatrix}\right),$$

(12)

where $\boldsymbol{\mu}_\mathbf{s}, \boldsymbol{\mu}_\mathbf{v}, \boldsymbol{\sigma}_\mathbf{s}, \boldsymbol{\sigma}_\mathbf{v}$ are encoding neural networks. The encoder for the initial position depends solely on the first item in the data sequence $\mathbf{x}_0$, whereas the encoder for the initial velocity depends on multiple data points $\mathbf{x}_{0:m}$, where $m \le N$ is the amortized inference length. We use neural network encoders and decoders whose architectures depend on the application (see the supplementary document for details). The variational approximation for the latent dynamics $q_{\text{ode}}(\mathbf{z}_{1:N}|\mathbf{x}_{0:N}, \mathbf{z}_0, \mathcal{W})$ is defined implicitly via the instantaneous change of variable for the second order ODEs shown in Eq. 5. The initial density is given by the encoder $q_{\text{enc}}(\mathbf{z}_0|\mathbf{x}_0)$, and density for later points can be solved by numerical integration using Eq. 6. Note that we treat the entire latent trajectory evaluated at observed time points, $Z \equiv \mathbf{z}_{0:N}$, as a latent variable, and the latent trajectory samples $\mathbf{z}_{1:N}$ are solved conditioned on the ODE initial values $\mathbf{z}_0$ and BNN parameter values $\mathcal{W}$. Finally, evidence lower bound (ELBO) becomes as follows (for brevity we define $X \equiv \mathbf{x}_{0:N}$):

$$\log p(X) \ge \underbrace{-\,\text{KL}[q(\mathcal{W}, Z|X)||p(\mathcal{W}, Z)] + \mathbb{E}_{q(\mathcal{W}, Z|X)}[\log p(X|\mathcal{W}, Z)]}_{\text{ELBO}}$$

(13)

$$= -\mathbb{E}_{q(\mathcal{W}, Z|X)}\left[\log \frac{q(\mathcal{W})q(Z|\mathcal{W}, X)}{p(\mathcal{W})p(Z)}\right] + \mathbb{E}_{q(\mathcal{W}, Z|X)}[\log p(X|\mathcal{W}, Z)]$$

(14)

$$= -\,\text{KL}[q(\mathcal{W})||p(\mathcal{W})] + \mathbb{E}_{q(\mathcal{W}, Z|X)}\left[-\log \frac{q(Z|\mathcal{W}, X)}{p(Z)} + \log p(X|\mathcal{W}, Z)\right]$$

(15)

$$= \underbrace{-\,\text{KL}[q(\mathcal{W})||p(\mathcal{W})]}_{\text{ODE regularization}} + \underbrace{\mathbb{E}_{q_{\text{enc}}(\mathbf{z}_0|X)}\left[-\log \frac{q_{\text{enc}}(\mathbf{z}_0|X)}{p(\mathbf{z}_0)} + \log p(\mathbf{x}_0|\mathbf{z}_0)\right]}_{\text{VAE loss}}$$

$$+ \sum_{i=1}^{N} \underbrace{\mathbb{E}_{q_{\text{ode}}(\mathcal{W}, \mathbf{z}_i | X, \mathbf{z}_0)} \left[ -\log \frac{q_{\text{ode}}(\mathbf{z}_i | \mathcal{W}, X)}{p(\mathbf{z}_i)} + \log p(\mathbf{x}_i | \mathbf{z}_i) \right]}_{\text{dynamic loss}} \qquad (16)$$

where the prior distribution $p(\mathcal{W}, \mathbf{z}_0)$ is a standard Gaussian. The prior density follows Eq. 6 with $\mathbf{f}_{\mathcal{W}}$ replaced by the unknown $\mathbf{f}_{\text{true}}$, which causes $p(\mathbf{z}_t)$, $t > 1$ to be intractable.[2] Thus, we resort to a simplifying assumption and place a standard regularizing Gaussian prior over $\mathbf{z}_{1:N}$.

We now examine each term in Eq. 16. The first term is the BNN weight penalty, which helps avoiding overfitting. The second term is the standard VAE bound, meaning that VAE is retrieved for sequences of length 1. The only (but major) difference between the second and the third terms is that the expectation is computed with respect to the variational distribution induced by the second order ODE. Finally, we optimize the Monte Carlo estimate of Eq. 16 with respect to variational posterior $\{\mathbf{m}, s\}$, encoder and decoder parameters, and also make use of reparameterization trick to tackle uncertainties in both the initial latent states and in the acceleration dynamics (Kingma and Welling, 2013).

### 3.3 Penalized variational loss function

A well-known pitfall of VAE models is that optimizing the ELBO objective does not necessarily result in accurate inference (Alemi et al., 2017). Several recipes have already been proposed to counteract the imbalance between the KL term and reconstruction likelihood (Zhao et al., 2017; Higgins et al., 2017). In this work, we borrow the ideas from Higgins et al. (2017) and weight the $\text{KL}[q(\mathcal{W})||p(\mathcal{W})]$ term resulting from the BNN with a constant factor $\beta$. We choose to fix $\beta$ to the ratio between the latent space dimensionality and number of weight parameters, $\beta = |q|/|\mathcal{W}|$, in order to counter-balance the penalties on latent variables $\mathcal{W}$ and $\mathbf{z}_i$.

Our variational model utilizes encoders only for obtaining the initial latent distribution. In cases of long input sequences, dynamic loss term can easily dominate VAE loss, which may cause the encoders to underfit. The underfitting may also occur in small data regimes or when the distribution of initial data points differs from data distribution. In order to tackle this, we propose to minimize the distance between the encoder distribution and the distribution induced by the ODE flow (Eqs. 12 and 6). At the end, we have an alternative, penalized target function, which we call ODE$^2$VAE-KL:

$$\mathcal{L}_{\text{ODE}^2\text{VAE}} = -\beta \, \text{KL}[q(\mathcal{W})||p(\mathcal{W})] + \mathbb{E}_{q(\mathcal{W}, Z | X)} \left[ -\log \frac{q(Z | \mathcal{W}, X)}{p(Z)} + \log p(X | \mathcal{W}, Z) \right] \qquad (17)$$
$$- \gamma \mathbb{E}_{q(\mathcal{W})} \left[ \text{KL}[q_{\text{ode}}(Z | X)||q_{\text{enc}}(Z | \mathcal{W}, X)] \right].$$

We choose the constant $\gamma$ by cross-validation. In practice, we found out that an annealing scheme in which $\gamma$ is gradually increased helps optimization, which is also used in (Karl et al., 2016; Rezende and Mohamed, 2015).

### 3.4 Related work

Despite the recent VAE and GAN breakthroughs, little attention has been paid to deep generative architectures for sequential data. Existing VAE-based sequential models rely heavily on RNN encoders and decoders (Chung et al., 2015; Serban et al., 2017), with very few interest in stochastic models (Fraccaro et al., 2016). Some research has been carried out to approximate latent dynamics by LSTMs (Lotter et al., 2016; Hsu et al., 2017; Li and Mandt, 2018), which results in observations to be included in latent transition process. Consequently, the inferred latent space and dynamics do not fully reflect the observed phenomena and usually fail to produce decent long term predictions (Karl et al., 2016). In addition, RNNs are shown to be incapable of accurately modeling nonuniformly sampled sequences (Chen et al., 2018b), despite the recent efforts that incorporate time information in RNN architectures (Li et al., 2017; Xiao et al., 2018).

Recently, neural ODEs introduced learning ODE systems with neural network architectures, and proposed it for the VAE latent space as well for simple cases (Chen et al., 2018b). In Gaussian process prior VAE, a GP prior is placed in the latent space over a sequential index (Casale et al., 2018). To the best of our knowledge, there is no work connecting second order ODEs and Bayesian neural networks with VAE models.

Table 1: Comparison of VAE-based models

| Method | Higher order | Continuous-time | Stochastic dynamics | state | Reference |
|---|---|---|---|---|---|
| VAE | ✗ | ✗ | ✗ | ✓ | Kingma and Welling (2013) |
| VRNN | ✗ | ✗ | ✗ | ✓ | Chung et al. (2015) |
| SRNN | ✗ | ✗ | ✓ | ✓ | Fraccaro et al. (2016) |
| GPPVAE | ✗ | ✓* | ✗ | ✓ | Casale et al. (2018) |
| DSAE | ✗ | ✗ | ✓ | ✓ | Li and Mandt (2018) |
| Neural ODE | ✗ | ✓ | ✗ | ✓ | Chen et al. (2018b) |
| ODE$^2$VAE | ✓ | ✓ | ✓ | ✓ | current work |

* GPPVAE uses a latent GP prior but only a discrete case was demonstrated in Casale et al. (2018).

## 4   Experiments

We illustrate the performance of our model on three different datasets: human motion capture (see the acknowledgements), rotating MNIST (Casale et al., 2018) and bouncing balls (Sutskever et al., 2009). Our goal is twofold: First, given a walking or bouncing balls sequence, we aim to predict the future sensor readings and frames. Second, we would like to interpolate an unseen rotation angle from a sequence of rotating digits. The competing techniques are specified in each section. For all methods, we have directly applied the public implementations provided by the authors. Also, we have tried several values for the hyper-parameters with the same rigor and we report the best results. To numerically compare the models, we sample 50 predictions per test sequence and report the mean and standard deviation of the mean squared error (MSE) over future frames. We include the mean MSE of mean predictions (instead of trajectory samples) in the supplementary.

We implement our model in Tensorflow (Abadi et al., 2016). Encoder, differential function and the decoder parameters are jointly optimized with Adam optimizer (Kingma and Ba, 2014) with learning rate 0.001. We use Tensorflow's own `odeint_fixed` function, which implements fourth order Runge-Kutta method, for solving the ODE systems on a time grid that is five times denser than the observed time points. Neural network hyperparameters, chosen by cross-validation, are detailed in the supplementary material. We also include ablation studies with deterministic NNs and first order dynamics in the appendix.

### 4.1   CMU walking data

To demonstrate that our model can capture arbitrary dynamics from noisy observations, we experiment on two datasets extracted from CMU motion capture library. First, we use the dataset in Heinonen et al. (2018), which consists of 43 walking sequences of several subjects, each of which is fitted separately. The first two-third of each sequence is reserved for training and validation, and the rest is used for testing. Second dataset consists of 23 walking sequences of subject 35 (Gan et al., 2015), which is partitioned into 16 training, 3 validation and 4 test sequences. We followed the preprocessing described in Wang et al. (2008), after which we were left with 50 dimensional joint angle measurements.

We compare our ODE$^2$VAE against a GP-based state space model GPDM (Wang et al., 2008), a dynamic model with latent GP interpolation VGPLVM (Damianou et al., 2011), two black-box ODE solvers npODE (Heinonen et al., 2018) and neural ODEs (Chen et al., 2018b), as well as an RNN-based deep generative model DTSBN-S (Gan et al., 2015). In test mode, we input the first three frames and the models predict future observations. GPDM and VGPLVM are not applied to the second dataset since GPDM optimizes its latent space for input trajectories and hence does not allow simulating dynamics from any random point, and VGPLVM implementation does not support multiple input sequences.

The results are presented in Table 2. First, we reproduce the results in Heinonen et al. (2018) by obtaining the same ranking among GPDM, VGPLVM and npODE. Next, we see that DTSBN-S is not able to predict the distant future accurately, which is a well-known problem with RNNs. As expected, all models attain smaller test errors on the second, bigger dataset. We observe that neural ODE usually perfectly fits the training data but failed to extrapolate on the first dataset. This overfitting problem

Table 2: Average MSE on future frames

| | Test error | | |
|---|---|---|---|
| Model | Mocap-1 | Mocap-2 | Reference |
| GPDM | $126.46 \pm 34$ | N/A | Wang et al. (2008) |
| VGPLVM | $142.18 \pm 1.92$ | N/A | Damianou et al. (2011) |
| DTSBN-S | $80.21 \pm 0.04$ | $34.86 \pm 0.02$ | Gan et al. (2015) |
| NPODE | 45.74 | 22.96 | Heinonen et al. (2018) |
| NEURALODE | $87.23 \pm 0.02$ | $22.49 \pm 0.88$ | Chen et al. (2018b) |
| ODE$^2$VAE | $93.07 \pm 0.72$ | $10.06 \pm 1.4$ | current work |
| ODE$^2$VAE-KL | $\mathbf{15.99 \pm 4.16}$ | $\mathbf{8.09 \pm 1.95}$ | current work |

is not surprising considering the fact that only ODE initial value distribution is penalized. On the contrary, our ODE$^2$VAE regularizes its entire latent trajectory and also samples from the acceleration field, both of which help tackling overfitting problem. We demonstrate latent state trajectory samples and reconstructions from our model in the supplementary.

## 4.2 Rotating MNIST

Next, we contrast our ODE$^2$VAE against recently proposed Gaussian process prior VAE (GPPVAE) (Casale et al., 2018), which replaces the commonly *iid* Gaussian prior with a GP and thus performs latent regression. We repeat the experiment in Casale et al. (2018) by constructing a dataset by rotating the images of handwritten "3" digits. We consider the same number of rotation angles (16), training and validation sequences (360&40), and leave the same rotation angle out for testing (see the first row of Figure 4b for the test angle). In addition, four rotation angles are randomly removed from each rotation sequence to introduce non-uniform sequences and missing data (an example training sequence is visualized in the first row of Figure 4a).

Test errors on the unseen rotation angle are given in Table 3. During test time, GPPVAE encodes and decodes the images from the test angle, and the reconstruction error is reported. On the other hand, ODE$^2$VAE only encodes the first image in a given sequence, performs latent ODE integration starting from the encoded point, and decodes at given time points - *without* seeing the test image even in test mode. In that sense, our model is capable of generating images with arbitrary rotation angles. Also note that both models make use of the angle/time information in training and test mode. An example input sequence with missing values and corresponding reconstructions are illustrated in Figure 4a, where we see that ODE$^2$VAE nicely fills in the gaps. Also, Figure 4b demonstrates our model is capable of accurately learning and rotating different handwriting styles.

Table 3: Average prediction errors on test angle

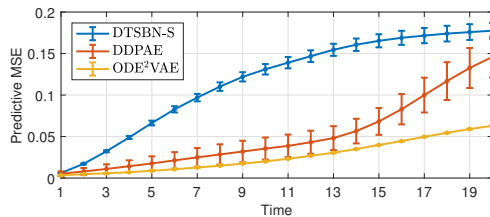

Figure 3: Bouncing balls errors.

| MODEL | TEST ERROR |
|---|---|
| GPPVAE-DIS$^\diamond$ | $0.0309 \pm 0.00002$ |
| GPPVAE-JOINT$^\diamond$ | $0.0288 \pm 0.00005$ |
| ODE$^2$VAE | $0.0194 \pm 0.00006$ |
| ODE$^2$VAE-KL | $\mathbf{0.0188 \pm 0.0003}$ |

## 4.3 Bouncing balls

As a third showcase, we test our model on bouncing balls dataset, a standard benchmark used in generative temporal modeling literature (Gan et al., 2015; Hsieh et al., 2018; Lotter et al., 2015). The dataset consists of video frames of three balls bouncing within a rectangular box and also colliding with each other. The exact locations of the balls as well as physical interaction rules are to be inferred from the observed sequences. We make no prior assumption on visual aspects such as ball count, mass, shape or on the underlying physical dynamics.

**(a) A training sequence and its reconstruction**

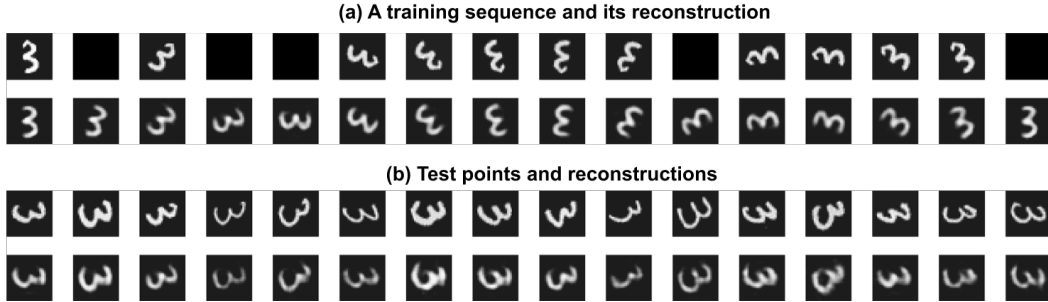

**(b) Test points and reconstructions**

Figure 4: Panel **(a)** shows a training sequence with missing values (first row) and its reconstruction (second row). First row in panel **(b)** demonstrates test angles from different sequences, i.e., handwriting styles, and below are model predictions.

We have generated a training set of 10000 sequences of length 20 frames and a test set of 500 sequences using the implementation provided with Sutskever et al. (2009). Each frame is 32x32x1 and pixel values vary between 0 and 1. We compare our method against DTSBN-S (Gan et al., 2015) and decompositional disentangled predictive auto-encoder (DDPAE) (Hsieh et al., 2018), both of which conduct experiments on the same dataset. In test mode, first three frames of an input sequence are given as input and per pixel MSE on the following 10 frames are computed. We believe that measuring longer forecast errors is more informative about the inference of physical phenomena than reporting one-step-ahead prediction error, which is predominantly used in current literature (Gan et al., 2015; Lotter et al., 2015).

Predictive errors and example reconstructions are visualized in Figures 3 and 5. The RNN-based DTSBN-S nicely extrapolates a few frames but quickly loses track of ball locations and the error escalates. DDPAE achieves a much smaller error over time; however, we empirically observed that the reconstructed images are usually imperfect (here, generated balls are bigger than the originals), and also the model sometimes fails to simulate ball collisions as in Figure 5. Our ODE$^2$VAE generates long and accurate forecasts and significantly improves the current state-of-the-art by almost halving the error. We empirically found out that a CNN encoder that takes channel-stacked frames as input yields smaller prediction error than an RNN encoder. We leave the investigation of better encoder architectures as an interesting future work.

## 5 Discussion

We have presented an extension to VAEs for continuous-time dynamic modelling. We decompose the latent space into position and velocity components, and introduce a powerful neural second order differential equation system. As shown empirically, our variational inference framework results in Bayesian neural network that helps tackling overfitting problem. We achieve state-of-the-art performance in long-term forecasting and imputation of high-dimensional image sequences.

There are several directions in which our work can be extended. Considering divergences different than KL would lead to Wasserstein auto-encoder formulations (Tolstikhin et al., 2017). The latent ODE flow can be replaced by stochastic flow, which would result in an even more robust model. Proposed second order flow can also be combined with generative adversarial networks to produce real-looking videos.

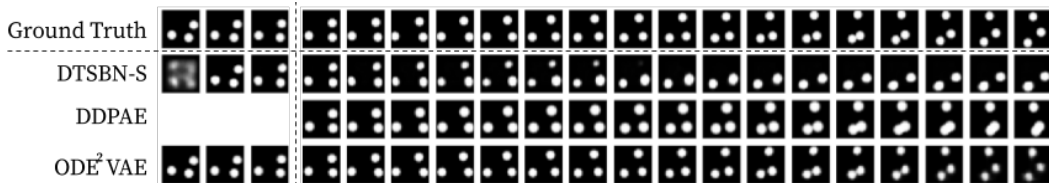

Figure 5: An example test sequence from bouncing ball experiment. Top row is the original sequence. Each model takes the first three frames as input and predicts the further frames.

**Acknowledgements.**

The data used in this project was obtained from `mocap.cs.cmu.edu`. The database was created with funding from NSF EIA-0196217. The calculations presented above were performed using computer resources within the Aalto University School of Science Science-IT project. This work has been supported by the Academy of Finland grants no. 311584 and 313271.

## Footnotes

[1]Time-dependent differential functions $\mathbf{f}(\mathbf{z}, t)$ can indirectly approximate higher-order dynamics.

[2]Although our variational approximation model assumes deterministic second-order dynamics, the underlying true model may also have more complex or stochastic dynamics.

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
