[Supplementary Material · ODE2VAE_supp.pdf]

# Supplementary Material for ODE$^2$VAE: Deep generative second order ODEs with Bayesian neural networks

**Çağatay Yıldız[1], Markus Heinonen[1,2], Harri Lähdesmäki[1]**
Department of Computer Science
Aalto University, Finland, FI-00076
{cagatay.yildiz, markus.o.heinonen, harri.lahdesmaki}@aalto.fi

## 1 Ablation studies

**1st-order baseline:** We tested a new ODE$^1$VAE variant where the latent space is governed by 1st-order ODE system. ODE$^1$VAE is similar to the NeuralODE Chen et al. (2018), except for having BNNs, and for NeuralODE placing a variational distribution on initial value $q(\mathbf{x}_0)$, while ODE$^1$VAE models the posterior over full trajectory $q(\mathbf{x}_{0:T})$.

**ODE$^1$VAE vs ODE$^2$VAE:** We performed a new comparison study of ODE$^1$VAE against ODE$^2$VAE on bouncing balls dataset. The experimental setup is kept the same, except that the number of convolutional filters is reduced so that the impact of differential function choice becomes more apparent. Table 1 shows the resulting MSE over 10 frame ahead predictions. Note that ODE$^2$VAE models the acceleration $\dot{\mathbf{v}}_t = \mathbf{f}(\mathbf{s}_t, \mathbf{v}_t) : \mathbb{R}^{2d} \to \mathbb{R}^d$ whereas 1st-order systems learn $\dot{\mathbf{z}}_t = \mathbf{f}(\mathbf{z}_t) : \mathbb{R}^d \to \mathbb{R}^d$. Results show that the 2nd-order dynamics results in far better accuracy, even if the first order dynamics has more flops ($d = 50$). We will include ablation studies in the paper.

**NN vs BNN:** Table 1 shows comparable performance of BNNs and NNs on bouncing balls. In order to demonstrate the benefit of using a BNN, we repeat the CMU walking experiment with a NN differential function. The MSE achieved by ODE$^2$VAE-NN over three test sequences is 9.96, whereas ODE$^2$VAE-BNN error improves to 9.43.

Table 1: Comparison of neural network (NN) and Bayesian neural network (BNN) ODE's with different latent dimensionalities on BOUNCING BALL experiment. Adding 2nd order momentum achieves superior performance, while BNN's have a smaller impact.

| Model | Latent dimensions $d$ | | Test MSE | |
|---|---|---|---|---|
| | 1st-order state | 2nd-order momentum | NN | BNN |
| ODE$^1$VAE | 25 | - | 45 | 43 |
| | 50 | - | 36 | 35 |
| ODE$^2$VAE | 25 | 25 | **26** | **27** |

## 2 Extra results

Below, we report the MSEs of mean trajectories, which are obtained with mean model predictions (e.g., for our model, when the mean value from the encoder distribution and variational posterior is used).

Table 2: Average mean MSE on future mocap frames

| Model | Test error | | Reference |
|---|---|---|---|
| | Dataset 1 | Dataset 2 | |
| GPDM | 57.52 | N/A | Wang et al. (2008) |
| VGPLVM | 128.03 | N/A | Damianou et al. (2011) |
| DTSBN-S | 78.39 | 37.20 | Gan et al. (2015) |
| NPODE | 45.74 | 22.96 | Heinonen et al. (2018) |
| NEURALODE | 97.74 | 21.60 | Chen et al. (2018) |
| ODE$^2$VAE | 32.19 | 17.20 | current work |
| ODE$^2$VAE-KL | **30.72** | **6.48** | current work |

Table 3: Mean prediction errors on test angle of rotating MNIST dataset ($\diamond$ taken from Casale et al. (2018))

| MODEL | TEST ERROR |
|---|---|
| GPPVAE-DIS$^\diamond$ | 0.0306 |
| GPPVAE-JOINT$^\diamond$ | 0.0280 |
| ODE$^2$VAE | 0.0204 |
| ODE$^2$VAE-KL | **0.0184** |

Figure 1: Mean prediction errors on bouncing balls dataset.

## 3 Experiment details

### 3.1 CMU mocap

We consider two different datasets. Here is a link to the first one (with 43 sequences) and here is a link to the second dataset. We set $\gamma = 1$. We tried out the architecture in Figure 4 with 1/2 hidden layers, 30/50 hidden units, tanh/relu/no activation functions. We found out that 2 hidden layers, 30 units and tanh performs the best. Each experiment is executed on a standard laptop for around 3 hours. The latent dimensionality is fixed to 6 for all models, i.e., $\mathbf{s}_t, \mathbf{v}_t \in \mathbb{R}^3$.

We visualize the position trajectories in Figure 2 for cases in which either encoder/BNN variational posteriors are sampled or the mean values are used. Note that latent field that is considered in our work corresponds to the right-most panel, whereas neural ODEs considers the second one.

Figure 2: Example latent trajectories from CMU mocap experiment

### 3.2 Rotating MNIST

Here is the dataset. We set $\gamma = 1$. We tried out 4/8/12 as the number of layers in the first layers of encoders and 8/12/16 as the last layer of the decoder. The code is executed on NVIDIA Tesla V100 GPUs for around 4 hours. The latent dimensionality is fixed to 16 for all models, i.e., $\mathbf{s}_t, \mathbf{v}_t \in \mathbb{R}^8$.

Figure 3: Comparison of our method against neural ODEs on CMU mocap data set. Each panel demonstrates a sensor measurement plotted over time.

Figure 4: CMU mocap walking data experiment neural architectures

## 3.3 Bouncing balls

Here is the dataset. We set $\gamma = 0.001$. We tried out 8/16/32 as the number of layers in the first layers of encoders and 16/32 as the last layer of the decoder. We also experimented with relu and tanh activations. The code is executed on NVIDIA Tesla V100 GPUs for around 3 days. The latent dimensionality is fixed to 50 for all models, i.e., $\mathbf{s}_t, \mathbf{v}_t \in \mathbb{R}^2 5$. Also note that we obtained the same error when $\mathbf{s}_t, \mathbf{v}_t \in \mathbb{R}^5 0$.

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

## a) Momentum Encoder

first three frames stacked
28x28x3

5x5 stride 2 + BN + relu
14x14x8

5x5 stride 2 + BN + relu
7x7x16

5x5 stride 2 + BN + relu
4x4x32

5x5 stride 2 + BN + relu
2x2x64

FC    FC

momentum    momentum
mean        variance
1x20        1x20

## b) Position Encoder

first frame
28x28x1

5x5 stride 2 + BN + relu
14x14x8

5x5 stride 2 + BN + relu
7x7x16

5x5 stride 2 + BN + relu
4x4x32

5x5 stride 2 + BN + relu
2x2x64

FC    FC

momentum    momentum
mean        variance
1x20        1x20

## c) Differential Function

position & momentum
1x40

FC-100 + tanh

FC-100 + tanh

FC

momentum differential
1x20

## d) Decoder

position
1x20

FC-72 (3x3x8)

3x3 stride 2 + BN + relu
7x7x48

5x5 stride 2 + BN + relu
14x14x24

5x5 stride 2 + BN + relu
28x28x12

5x5 stride 1 + sigmoid

reconstruction
28x28x1

Figure 5: Rotating MNIST experiment neural architectures

## a) Momentum Encoder

first three frames stacked
32x32x3

5x5 stride 2 + BN + relu
16x16x16

5x5 stride 2 + BN + relu
8x8x32

5x5 stride 2 + BN + relu
4x4x64

5x5 stride 2 + BN + relu
2x2x128

FC    FC

momentum    momentum
mean        variance
1x25        1x25

## b) Position Encoder

first frame
32x32x1

5x5 stride 2 + BN + relu
16x16x16

5x5 stride 2 + BN + relu
8x8x32

5x5 stride 2 + BN + relu
4x4x64

5x5 stride 2 + BN + relu
2x2x128

FC    FC

momentum    momentum
mean        variance
1x25        1x25

## c) Differential Function

position & momentum
1x50

FC-100 + tanh

FC-100 + tanh

FC

momentum differential
1x25

## d) Decoder

position
1x25

FC-128 (4x4x8)

3x3 stride 2 + BN + relu
8x8x128

5x5 stride 2 + BN + relu
16x16x64

5x5 stride 2 + BN + relu
32x32x32

5x5 stride 1 + sigmoid

reconstruction
32x32x1

Figure 6: Bouncing balls experiment neural architectures

Jack M Wang, David J Fleet, and Aaron Hertzmann. Gaussian process dynamical models for human motion. *IEEE transactions on pattern analysis and machine intelligence*, 30(2):283–298, 2008.