[Reviews · NeurIPS 2019]

Reviewer 1



Overall the paper is very interesting and proposes a novel method. However, there are questions about generalization of the approach beyond the simple datasets/tasks tested in the paper

Reviewer 2



Summary: The paper looks at the problem of modelling sequential data, specifically image data. It proposes to combine a (beta-)VAE model with a Neural ODE. The VAE encodes the input image to a location and velocity, the Neural ODE computes the dynamics over time, the VAE then decodes using the location parameters. To model the velocity, the authors extend the Neural ODE to be second order. The paper contains extensive introduction to the method, including ODE, VI, beta-VAE, generative models, ODE flow. The model used for the dynamics is a Bayesian Neural Network, which is a neural network with a distribution over every weight. The output of this model is a distribution. The authors show impressive results on the CMU walking data set, bouncing balls, and Rotating MNIST, comparing to a variety of different methods. Discussion: The paper is well written, and introduces an interesting application of the Neural ODE. The approach of embedding the Neural ODE within other approaches is particularly appealing and seems like a strong way of encoding inductive bias into the model. Section 2 and 3 are an excellent introduction into the multitude of components used in the model. Figure 1 and 2 are both great illustrations. The experiments are sensible and described in detail, including particular ways alternative methods were evaluated. The usage and efficacy of the Bayesian neural network is not well explained. It's unclear if using a BNN over a regular NN gave any advantage, further it's unclear how the BNN was optimized. The authors mention variational inference but do not go further into detail (reparametrization trick? mean field?). Did the authors find any difference in NFEs for solving the ODE between using a BNN vs NN? The paper only briefly explain ODE2VAE-KL, and the difference between (16) and (17) can use more explanation. Do you run the encoder for every triple of input frames and compare that with the output of the ODE at those times? i.e. how do I interpret the third term of (17)? Given that ODE2VAE-KL is the best performing model, it would help to explain why this setup is beneficial and perhaps an analysis into the difference with regular ODE2VAE. The paper compares with fairly dated method (aside from the newer ODE based methods). Could the authors comment on performance compared to a new method such as "Sequential Attend, Infer, Repeat: Generative Modelling of Moving Objects" (Kosiorek et al.)? Especially on the more challenging dataset moving mnist? -- Rebuttal -- The rebuttal clarified some important points. I have raised my score to a 7.

Reviewer 3



1. It's original that this paper proposes a second order ODE that allows modelling the latent dynamic ODE state decomposed as position and momentum. 2. It's original to connect second order ODEs and Bayesian neural networks with VAE models. 3. The paper is well written and organized. 4. The method proposed in this paper can be applied for high-dimensional sequential data and learn the embedding of high dimensional trajectories. 5. The proposed methods are evaluated on a diverse of datasets with state-of-the-art performance.

[Author Response · NeurIPS 2019]

We thank all reviewers for their constructive feedback and for their time in creating well thought out reviews.

Below we address all raised concerns, namely we perform ablation studies of adding (i) 2nd-order ODEs and (ii) BNNs;
(iii) address more complex experiments and comparisons; and (iv) discuss the role of the KL and regularisation.

**A new 1st-order baseline:** We tested a new $\text{ODE}^1\text{VAE}$ variant where the latent space is governed by 1st-order ODE
system. $\text{ODE}^1\text{VAE}$ is similar to the NeuralODE [Chen et al 2018], except for having BNNs, and for NeuralODE placing
a variational distribution on initial value $q(\mathbf{x}_0)$, while $\text{ODE}^1\text{VAE}$ models the posterior over full trajectory $q(\mathbf{x}_{0:T})$.

**[R1,R3] $\text{ODE}^1\text{VAE}$ vs $\text{ODE}^2\text{VAE}$:** We
performed a new comparison study of
$\text{ODE}^1\text{VAE}$ against $\text{ODE}^2\text{VAE}$ on bounc-
ing balls dataset. The experimental setup
is kept the same, except that the number
of convolutional filters is reduced so that
the impact of differential function choice
becomes more apparent. Table 1 shows the
resulting MSE over 10 frame ahead pre-
dictions. Note that $\text{ODE}^2\text{VAE}$ models the
acceleration $\dot{\mathbf{v}}_t = \mathbf{f}(\mathbf{s}_t, \mathbf{v}_t) : \mathbb{R}^{2d} \to \mathbb{R}^d$

Table 1: Comparison of neural network (NN) and Bayesian neural network
(BNN) ODE's with different latent dimensionalities on BOUNCING BALL
experiment. Adding 2nd order momentum achieves superior performance,
while BNN's have a smaller impact.

| Model | Latent dimensions $d$ | | Test MSE | |
|---|---|---|---|---|
| | 1st-order state | 2nd-order momentum | NN | BNN |
| $\text{ODE}^1\text{VAE}$ | 25 | - | 45 | 43 |
| | 50 | - | 36 | 35 |
| $\text{ODE}^2\text{VAE}$ | 25 | 25 | **26** | **27** |

whereas 1st-order systems learn $\dot{\mathbf{z}}_t = \mathbf{f}(\mathbf{z}_t) : \mathbb{R}^d \to \mathbb{R}^d$. Results show that the 2nd-order dynamics results in far better
accuracy, even if the first order dynamics has more flops ($d = 50$). We will include ablation studies in the paper.

**[R1,R2] NN vs BNN:** Table 1 shows comparable performance of BNNs and NNs on bouncing balls. In order to
demonstrate the benefit of using a BNN, we repeat the CMU walking experiment with a NN differential function. The
MSE achieved by $\text{ODE}^2\text{VAE}$-NN over three test sequences is 9.96, whereas $\text{ODE}^2\text{VAE}$-BNN error improves to 9.43.

**[R2] Learning of BNNs:** Learning BNN is performed via mean-field variational approximation (simultaneously with
variational inference of the whole $\text{ODE}^2\text{VAE}$ model), where each weight and bias component has its own mean and
shares a global variance parameter. The ODE solver used in our experiments is fixed step Runge-Kutta for both NN and
BNN systems; hence NFEs are also the same.

**[R1] Comprehensive experiments:** Our model is suitable for sequential datasets, of which we demonstrated good
performance on motion capture data, bouncing balls experiments and on rotating MNIST. Conventional image datasets
such as CIFAR-10 or Celeb are not directly applicable for our model as they do not have an immediate dynamic
dimension. In this work we proposed the theoretical foundations of latent differential equations, and in future we intend
to explore video prediction application as separate work due to its daunting scope and complexity.

**[R2] Comparison to moving MNIST:** Moving MNIST is a dataset of digits bouncing off the walls of a box. Physical
interaction rules in bouncing balls dataset is more complicated because balls collide with each other, as well. In that
sense, inferring the dynamics in bouncing balls dataset is more challenging. On the other hand, MNIST dataset possibly
requires more powerful decoders, which we will consider as part of future work on video prediction.

**[R3] Missing NeuralODE baseline in rotating MNIST and bouncing balls:** While the public NeuralODE imple-
mentation worked as expected in the CMU walking experiments, we were unable to get NeuralODE model to work in
BOUNCING BALLS and ROTATING MNIST datasets. We included ConvNet architectures and tried these experiments
numerous times with different encoder/decoder hyperparameters and initialisations; however we always got fully black
frames as reconstructions. We believe the $\text{ODE}^1\text{VAE}$ results instead to be informative enough to demonstrate inherent
limitations of 1st-order models, such as NeuralODE.

**[R1] Regularisation parameters:** The $\beta$ and $\gamma$ parameters weigh the regularising KL terms to be comparable to the
weight of the likelihood term (see e.g. "Fixing the Broken ELBO" paper). We choose to fix $\beta = |q|/|\mathbb{W}|$ to the ratio
between the latent space dimensionality $q$ and number of weight parameters of the differential function $|\mathbb{W}|$, in order to
counter-balance the penalties. We chose $\gamma = 0.001$ by cross validation from [0,0.1,0.01,...0.00001].

**[R2] $\text{ODE}^2\text{VAE}$-KL variant:** As correctly pointed out by the reviewer, all consecutive triplets in a sequence are
encoded. We then compute the KL divergence between encoder distributions and the state distributions induced by
ODE integration. This way, the entire sequence (rather than only the initial values) is utilized for encoder training.

**[R3] Long-term forecasting:** Long-term forecasting of non-linear dynamical systems requires an almost perfect
underlying dynamics model for the trajectories not to deviate. We regard "long-term" forecasting to be up around
20 frames ahead in bouncing balls, multiple cycles of walking, or a full rotation of MNIST numbers. We found out
empirically that NeuralODE can not forecast sufficiently, while the GPPVAE model interpolates states over time with
an RBF kernel with little extrapolation capability.

[Meta-Review · NeurIPS 2019]

This paper combine several modeling ingredients (BNNs, ODEs, and VAEs) to produce a new family of models. It's not clear to my whether adding second-order dynamics in particular is advantageous over just adding extra latent dimensions to the state, which I think would be a generalization of the current approach. However, seeing a comparison against GPLVM-based models was nice, since these two approaches represent very different technical approaches to the same problem.